



**Nitrophobic ectomycorrhizal fungi are associated with enhanced hydrophobicity of soil organic matter in a Norway spruce forest.**

[1]Juan Pablo Almeida, [1]Nicholas Rosenstock,, [2]Susanne Woche, [2]Georg Guggenberger and [1]Håkan Wallander

[1]Department of Biology, MEMEG, Lund University, 22362 Lund, Sweden

[2]Institute of Soil Science, Leibniz Universität Hannover, Herrenhäuser Str. 2, 30419 Hannover, Germany

*Correspondence to*: Juan Pablo Almeida (jpalmeidava@gmail.com)





**Abstract**
In boreal forests an important part of the photo assimilates are allocated belowground to
support ectomycorrhizal fungi (EMF) symbiosis. The production of EMF extramatrical
mycelium can contribute to carbon (C) sequestration in soils but the extent of this contribution
depends on the composition of the EMF community. Some species can decrease soil C stocks
by degrading soil organic matter (SOM) and certain species may enhance soil C stocks by
producing hydrophobic mycelia which can reduce the rate of SOM decomposition. To test
how EMF communities contribute to the development of hydrophobicity in SOM we
incubated sand-filled fungal-ingrowth meshbags amended with maize compost for one, two or
three growing seasons in non-fertilized and fertilized plots in a young Norway spruce (*Picea*
*abies*) forest. We measured hydrophobicity as determined by the contact angle, the C/N ratios
in the meshbags contents along with the amount of new C entering the meshbags from outside
(determined by C3 input to C4 substrate), and related that to the fungal community
composition. The proportion of EMF species increased over time to become the dominant
fungal guild after three growing seasons. Fertilization significantly reduced fungal growth and
altered EMF communities. In the control plots the most abundant EMF species was
*Piloderma oliviceum*, which was absent in the fertilized plots. The hydrophobicity of the
meshbag contents reached the highest values after three growing seasons only in the
unfertilized controls plots and was positively related to the abundance of *P. olivaceum,* the
C/N ratios of the meshbag contents, and the amount of new C in the meshbags. These results
suggest that some EMF species are associated with higher hydrophobicity of SOM and that
EMF community shifts induced by fertilization may result in reduced hydrophobicity of soil
organic matter which in turn may reduce C sequestration rates.
**Key words:** Ectomycorrhizal fungi, Contact angle, hydrophobicity, fertilization, fungal
25                  communities.





## 1 Introduction

Fertilization of forests has been suggested as a way to increase C sequestration to mitigate

climate change (Jörgenssen et al., 2021). In support for this, Bergh et al. (2008) found more

than doubling of aboveground growth of young Norway spruce forests in response to yearly

additions of a complete fertilizer in experimental sites in Sweden. A major part of gross

primary production, between 25% and 63% according to Litton et al. (2007), is however

allocated belowground to roots and associated ectomycorrhizal fungi, and this portion usually

declines in response to fertilization (Högberg, 2010). In support for this, reduced growth of

EMF mycelium was found in the young fertilized Norway spruce stands studied by Bergh et

al. (2008) (Wallander et al., 2011).

EMF form extensive mycelial networks, which efficiently distribute C in the soil

(Smith and Read, 2008), and this mycelium is turning into necromass when the mycelium

dies. Necromass from different EMF species decomposes at different rates (Koide et al.,

2009). Melanin content appears to have a negative influence for necromass decomposition,

but physical protection is also an important factor to reduce decomposition according to

Fernadez et al. (2016). SOM can be protected from decomposition in aggregates where

hydrophobic coatings of mineral particles limit water penetration (Goebel et al., 2011 ; von

Lützow et al., 2006), and hydrophobic SOM generally decomposes slower than hydrophilic

SOM (Nguyen and Harvey, 2003; 2001). Since some EMF species form hydrophobic, while

others form hydrophilic mycelia (Unestam and Sun, 1995), the composition of the EMF

community may thus have fundamental importance for the SOM properties and subsequently

for carbon sequestration rates in the soil.

In contrast to carbon accumulating activities by EMF, certain species may also reduce

soil C stocks by mineralizing nitrogen (N) and phosporus (P) from SOM  (Shah et al., 2016;

Lindahl and Tunlid, 2015; Bödecker et al., 2014). Bending and Read, 2015 demonstrated that



large amounts of N (23 %) and P (22%) in SOM can be mobilized and transferred to the host
plant in mesocoms grown in the laboratory, and this modification of SOM is likely to reduce
further decomposition performed by saprophytes in the soil (Fernandez and Kenedy, 2015 ;
Gadgil and Gadgil, 1971). Ectomycorrhizal fungi may thus have opposing effects on the
amount of SOM, and differences in community composition was proposed as one explanation
to different C accumulation rates in boreal forests in northern Sweden (Clemmensen et al.,
2015 ; 2013); later successional stages that accumulated more C were dominated by ericoid
mycorrhizal fungi with recalcitrant necromass, while younger successional stages that
accumulated less C were dominated by EMF of long distance exploration types with a high
capacity to degrade soil organic matter. Certain species of EMF may have exceptional
importance for organic matter accumulation as the presence of *Cortinarius acutus* resulted in
33% lower C storage in the organic top soils in 359 investigated stands in boreal forests in
Sweden (Lindahl et al., 2021).
It is well known that fertilization with N has a strong impact on growth and
composition of EMF (Lilleskov et al., 2011; Wallenda and Kottke 1988). Lilleskov et al.
(2011) demonstrated that species sensitive to N (e.g. *Cortinarius, Tricholoma Suillus, and*
*Piloderma*) usually produce hydrophobic mycelia while N tolerant species often produce
hydrophilic mycelia (e.g. *Laccaria, Russula, Lactarius*). Loss of hydrophobic EMF species at
high N input could thus have consequences for SOM formation and C sequestration rates, but
it is not well known to what extent EMF abundance has a significant effect on the overall
hydrophobicity of SOM.
In our study with young Norway spruce forests reported above (Wallander et al.,
2011), we used mesh bags amended with maize compost (C4 plant material enriched in $^{13}$C)
to estimate EMF fungal growth in control and fertilized plots. In the present study we
analysed the fungal communities as well as the hydrophobicity of the same mesh bag





contents. The mesh bags were harvested after one, two or three growing seasons in order to
follow fungal succession and development of hydrophobicity over time. All samples were
subjected to 454-sequencing in order to characterize the fungal communities. We expected
community composition to be influenced by fertilization, and hydrophobicity to increase over
time when EMF biomass and necromass accumulates. We also expected more N to be
removed by EMF from the mesh bags in the control than in the fertilized treatment. In
addition, we expected higher hydrophobicity in control versus fertilized plots due to a higher
proportion of hydrophobic species.

**2 Material and Method**
**2.1 Study site**
The experimental forest was located close to Ebbegärde in south-eastern Sweden (56°53'N
16°15E') in a 10 year old Norway spruce forest at time of sampling. The soil is a podzol on
coarse sandy glacial till (site index G29), and the depth of the humus layer varied between 3
and 8 cm.
The treatments were designed in randomized block design with 3 fertilization treatments and
3 blocks per treatment (n=3). The plot size was 40 x 40 m. The fertilization treatments were:
the unfertilized Control plots and 2 Fertilization regimes. The fertilization was applied by
hand as 50-100 kg N ha$^{-1}$ every year for the first fertilization regime and as 100-150 kg N ha$^{-1}$
every second year in the second fertilization regime. The supply of other macro- and
micronutrients was adjusted to initial target ratios of each element to N (Linder, 1995). For
this study both fertilization regimes were treated as one fertilization treatment. For a more
detailed description of the fertilization regime see Bergh et al. (2008) and Wallander et al.

99   (2011).




## 2.2 Experimental design

We used triangular shaped ingrowth bags made of nylon mesh (50 μm mesh size, 10 cm side,~1 cm thick) to capture fungi growing in the soil. This mesh size allows the ingrowth of fungal hyphae, but not roots (Wallander et al., 2001). The mesh bags were filled with 30 g acid-washed quartz sand 0.36-2.0 mm, 99.6% $SiO_2$, Ahlsell AB, Sweden) heated to a temperature of 600 °C overnight to remove all organic carbon. The sand was then mixed with 0.8% (w/w) maize compost. Maize compost was used since it has a unique C isotopic signature, which makes it possible to estimate C influx into the mesh bags. Results from these measurements are presented in Wallander et al. (2011), Maize compost was produced by cutting maize leaves into small pieces and compositing in an isolated plastic compost bin for 12 months. After that the compost was kept at +4 °C. Fresh compost was forced through a 2 mm mesh and then mixed with dry sand to make a uniform mixture.  The sand maize mixture had a carbon content of 0.4%. The bags were buried at approximately 5 cm depth in the interface between the organic horizon and the mineral soil where EM fungi are abundant (Lindahl et al., 2007). First harvest was done in November 2007, after 8 months incubation. The second harvest was done in November 2008 and the third harvest was done in November 2009. Four meshbags were pooled to make 1 composite sample for each block, year and treatment.  In the laboratory the mesh bags were opened and the contents from the four replicate mesh bags from each experimental plot were carefully pooled and mixed. Subsamples were taken for subsequent analyses (ergosterol, hydrophobicity, C and N content, fungal community) and immediately frozen.

The abundance of $\delta^{13}C$ as well as total C and N content were analyzed using an elemental analyzer (model EuroEA3024; Eurovector, Milan, Italy) connected to an Isoprime isotope-ratio mass spectrometer (Isoprime, Manchester, UK) as described by Wallander et al. (2011). The isotopic shift that occurred when $^{13}C$ depleted C (mainly EMF mycelia) entered the bags





from outside was used to calculate the amount of new C in the mesh bags. For details see
(Wallander et al. 2011).

**2.3 DNA extraction, PCR and 454 sequencing**
Ten grams of the sand/maize mixture from the composite samples was homogenized using a
ball mill without a ball (Retsch, Haan, Germany). DNA was extracted from the homogenized
samples by adding CTAB buffer (2 % cetyltrimetylammoniumbromid, 2 mM EDTA, 150 mM
Tris-HCl, pH 8), vortexing, and then incubating at 65 °C for 1.5 h, followed by chloroform
addition, vortexing, supernatant removal, and isopropanol and ethanol precipitation. The
pellet was resuspended in 50 µl of MiliQ-water (Millipore) and further cleaned using Wizard
DNA clean-up kit (Promega, Madison, WI, USA).
PCR was carried out for each sample in 3 triplicate 25 µl reactions, using the fungal-
specific primers ITS1-F (Gardes and Bruns 1993) and ITS4 (White et al. 1990). Each primer
was elongated with adaptors required for 454 pyrosequencing (ITS1-F/A adaptor and ITS4/B
adaptor). The ITS4 also contained a sample specific tag consisting of 8 bases; ITS1-F/A : 5´-
CCTATCCCCTGTGTGCCTTGGCAGTCTCAGCTTGGTCATTTAGAGGAAGTAA-3´;
ITS4/B :5´-
CCATCTCATCCCTGCGTGTCTCCGACTCAG*XXXXXXXX*TCCTCCGCTTATTGATATG
C-3´. PCR products were purified with Agencourt AMPure kit (Agencourt Bioscence
Corporation, Beverly, MA, USA) in order to remove residual salts, primers and primer
dimers. The concentration of the purified PCR products was measured with the PicoGreen ds
DNA Quantification Kit (Molecular Probes, Eugene, OR, USA) on a FLUOstar OPTIMA
(BMG LABTECH Gmbh, Ortenberg, Germany). Equal amounts of DNA from each sample
were pooled into one single pool and submitted for 454 pyrosequencing. Sequencing was



performed on a FLX 454 (Roche Applied Biosystems, City, Country) using the Lib-L
chemistry at the Pyrosequencing facility at Lund University, Lund, Sweden.

**2.4 Bioinformatic analysis**
After sequencing sequences were trimmed and filtered using Mothur v1.34 (Schloss et
al., 2009). Sequences outside the *ITS2* region and chimeric sequences were removed using
ITSx extractor v1.5.0 (Bengtsson-Palme et al., 2013). After filtering, a Bayesian clustering
was applied to the sequences using the Gaussian Mixture model CROP (Hao et al., 2011) at
97% sequence similarity, and a set of operational taxonomic units (OTUs) was thus obtained.
Clusters that were only found in one mesh bag sample (one PCR reaction) were excluded,
further reducing the possibility that any chimeric sequences were used in our analysis. Search
for sequence identities were performed by iteratively BLASTing (Basic Local Alignment
Search Tool) against 2 different sequence databases, the first was the UNITE (Koljalg et al.,
2005, http://unite.ut.ee/index.php) reference/representative sequence database (21,000 seqs,
dynamic taxa threshold, release date 2014-02-09), and the second was the full UNITE+INSD
sequence database (377,000 seqs, dynamic threshold, release date 2014-02-15)(Karsch-
Mizrachi et al., (2012). The UNITE and INSD databases were purged of all sequences, nearly
25% of the total, that did not have any taxonomic information, primarily environmental
samples from soils and roots using boolean terms (ex. Environmental, uncultured, root
endophyte, unidentified). Sequences were assigned to species when there was at least 97 %
similarity between query sequence and top hit. Sequences that failed to match at this threshold
were excluded. Separate clusters that matched the same database sequence were subsequently
lumped into one OTU.
Using names and taxonomy associated with the OTU's, the total fungal community was
divided by both phylum (Basidiomycota, Ascomycota, Zygomycota, and Chytridiomycota)



and function (known ectomycorrhizal fungi, unknown ectomycorrhizal status, saprotrophic
fungi); OTUs were considered known ectomycorrhizal fungi based on the knowledge of the
ecology of known close relatives (genera or below) according to Tedersoo et al. (2010).
After filtering, each sample was rarified to the median number of reads using the
"rrarefy" function in the VEGAN package (Oksanen et al., 2013) in R (R Core Team, 2013).
For community comparison (total, or for ectomycorrhizal fungi), all read abundances were
converted to fractional abundance, such that the read abundances for all OTUs for each
sample totaled to 1.

**2.5 Hydrophobicity**
The hydrophobicity was evaluated in terms of contact angle (CA) with the sessile drop
method (Bachmann et al., 2003), using a CCD-equipped CA microscope (OCA 15,
DataPhysics, Filderstadt, Germany). Here the angle a drop of water forms at the <solid-liquid-
vapor interphase is measured. This contact angle is used to describe the wettability of the
surface; a CA≥90 indicates a hydrophobic and a zero CA a hydrophilic surface. A CA>0° and
<90° indicates subcritical water repellency.
For measurement, material from the meshbags contents was fixed on a glass slide with
double-sided adhesive tape in an ideally one-grain layer. Placement of a water drop is
recorded and the initial CA evaluated after ending of mechanical disturbances by drop shape
analysis (ellipsoidal fit) and fitting tangents on the left and right side of the drop, using the
software SCA 20 (DataPhysics, Filderstadt, Germany; Goebel et al., 2013). CA is given as the
mean CA of the left and right side of the drop. As an estimate about CA stability, CA again
was evaluated after 1 s (denoted as $CA_{1s}$) and after 5 s (denoted as $CA_{5s}$; Bachmann *et al.,*

198 2021).



Three replicates from each treatment (Control or Fertilized) and each incubation period (2007,
2008, 2009) were used in the measurements. One slide per replicate was prepared and for
each slide six drops were placed and averaged to obtain one CA per replicate (n=6). Two
slides containing the non-incubated sand-maize compost mix were also analyzed as a non-
treated reference material. Due to the coarse texture of the meshbag material, the drop volume
was 6 µL.
**2.6 Statistical analysis**
The statistical analyses for the fungal communities were performed using the VEGAN
package (Oksanen et al., 2013) in R (R Core Team, 2013). Fungal communities were
visualized with ordination using non-parametric multidimensional scaling (NMDS) using the
metaMDS function. Differences in community structure were visually compared with
centroids and the associated 95 % confidence interval associated with a t-distribution around
the standard error of the centroid.  To detect if the fungal communities were significantly
influenced by the treatments (fertilization and incubation periods), permutational multivariate
analysis of variance (PERMANOVA; Anderson, 2014) was performed. Pairwise comparisons
between treatments were tested using pairwise Adonis test.
To test for differences in hydrophobicity (contact angle), C/N ratios, new C inside the
meshbags and ergosterol ANOVA and two ways ANOVA were performed using the CAR
package (Fox & Weisberg, 2019) in R (R Core Team, 2013). To test for differences in the
relative abundance of EMF species between the treatments, Dunn's test for non-parametrical
samples was performed (Dinno, 2015).

Principal component analysis (PCA) was used to analyze the relationships between the most
abundant fungal species and the properties of the meshbag contents (hydrophobicity (contact





angle), C/N ratios, new carbon inside the meshbags, ergosterol) using the package
FactoMineR (Lê et al., 2008) in R (R Core Team, 2013).

**3 Results**
**3.1 Fungal biomass**
The concentration of ergosterol, as an estimate of fungal biomass, in the mesh bags have been
reported earlier (Wallander et al., 2011) and is summarized in Table 1. In brief, ergosterol
content increased from a starting value of 0.7 (original maize compost) to 2.2 mg g$^{-1}$ in the
mesh bags after incubation for one growing season in control plots. After this the
concentration did not change significantly over the coming two years. In fertilized plots the
concentration was significantly lower than the control plots (ANOVA, F= 13.4; p<0.01)
Table 1:
Average and standard error of the ergosterol concentrations, total C%, C/N ratio, amount of new carbon (C3
mainly from EMF), % of EMF DNA reads, and contact angle determined 5 seconds after placement of water
droplets placed on mesh bags material amended with maize compost (CA$_{5S}$; estimation of contact angle stability)

| Treatment | Incubation time (years) | Ergosterol µg g$^{-1}$ | C (%) | C/N | Amount of new C mg g$^{-1}$ | % of EMF reads | CA$_{5s}$ °/*SD* |
|---|---|---|---|---|---|---|---|
| Initial material | | 0.7 | | 11.9 | | | 37.3±0.1 |
| Control | 1 | 2.2±0.5 | 0.38±0.02 | 13.2±0.5 | 0.6 ± 0.2 | 11.3±2.2 | 62±2.8 |
| Control | 2 | 2.3±0.3 | 0.43±0.11 | 14.3±0.4 | 0.9 ± 0.2 | 24,4±2.3 | 67±4.4 |
| Control | 3 | 1.8±0.1 | 0.42±0.07 | 14.6±0.3 | 1± 0.4 | 78.3±1.4 | 78±7 |
| Fertilized | 1 | 1.1±0.5 | 0.42±0.02 | 13 ±0.5 | 0.5±0.2 | 7 ±3.6 | 62±3.6 |
| Fertilized | 2 | 1.6±0.6 | 0.4±0.04 | 13 ±0.6 | 0.3±0.2 | 31.3±11.9 | 57±1.4 |
| Fertilized | 3 | 1.1±0.2 | 0.42±0.04 | 12.4±0.2 | 1 ± 0.3 | 71.8±6.3 | 53±7.5 |



**3.2 Hydrophobicity, C content and C/N ratio of SOM**
Incubation in the field significantly increased hydrophobicity of the meshbag contents in the
unfertilized control plots as indicated by $CA_{1s}$ and $CA_{5s}$ (ANOVA, F= 6.2; p<0.05 and
ANOVA, F=10.2; p<0.01; respectively). CA of the control plots increased with incubation
time, but only the $CA_{1s}$ and $CA_{5s}$ were significantly different from the reference material (non-
incubated sand-maize compost mix), i.e., stability of CA was increased (Fig 1a).
Incubation in the field also affected hydrophobicity of the meshbag contents in the fertilized
plots as indicated by the initial CA and $CA_{1s}$ and $CA_{5s}$ (ANOVA, F= 5.2; p<0.05; ANOVA,
F= 4.1; p=0.06 and ANOVA, F=3.9; p=0.05; respectively). The CA stability ($CA_{1s}$ and $CA_{5s}$)
was increased compared to the reference material only in the one-year incubation meshbags.
As time of incubation in the soil increased, however, CA decreased. After 3 years of
incubation the initial CA became significantly smaller in comparison with the reference
material (Fig 1b). There were significant differences in the CA (initial, $CA_{1s}$ and $CA_{5s}$)
between meshbags from the control and fertilized plots in the 3-years incubation bags with
smaller CA (initial, $CA_{1s}$ and $CA_{5s}$) for the fertilized plots compared to the control (2009; Fig
1c, ANOVA, F= 3.2; p<0.05; F= 3,1; p=0.05 and F=2.8; p=0.06; respectively), but not for the
first and second incubation year (2007, 2008)

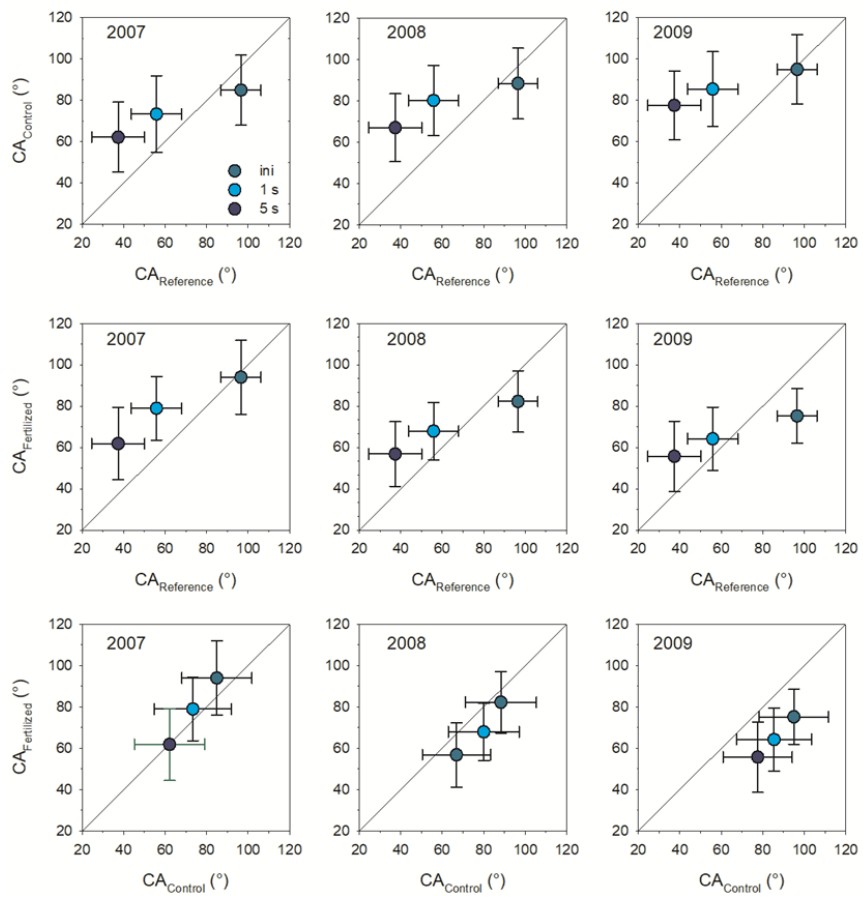


Figure 1: Comparison between the averaged contact angle (CA) (n=3) between a) control treatment and reference material b) fertilization treatment and reference material and c) control and fertilized treatments. Bars represent standard deviation.


262        The concentration of C in the mesh bags was not influenced by time or fertilization but

the amount of new C (C3-C presumably from EMF) in the mesh bags was significantly
affected by fertilization and was higher in the control plots than in the fertilized plots
according to the two-ways ANOVA (F=5.3 ; $p < 0.05$). The amount of new C tended to increase
with incubation time in the control plots (Table 1). The interaction between fertilization and
incubation time were not significant.



There was a positive correlation between the amount of new C and the hydrophobicity
for both $CA_{1s}$ and $CA_{5s}$ (Pearson, T = 2 , p=0.06; T = 1.9,  p=0.07 ; respectively) (Fig 2)

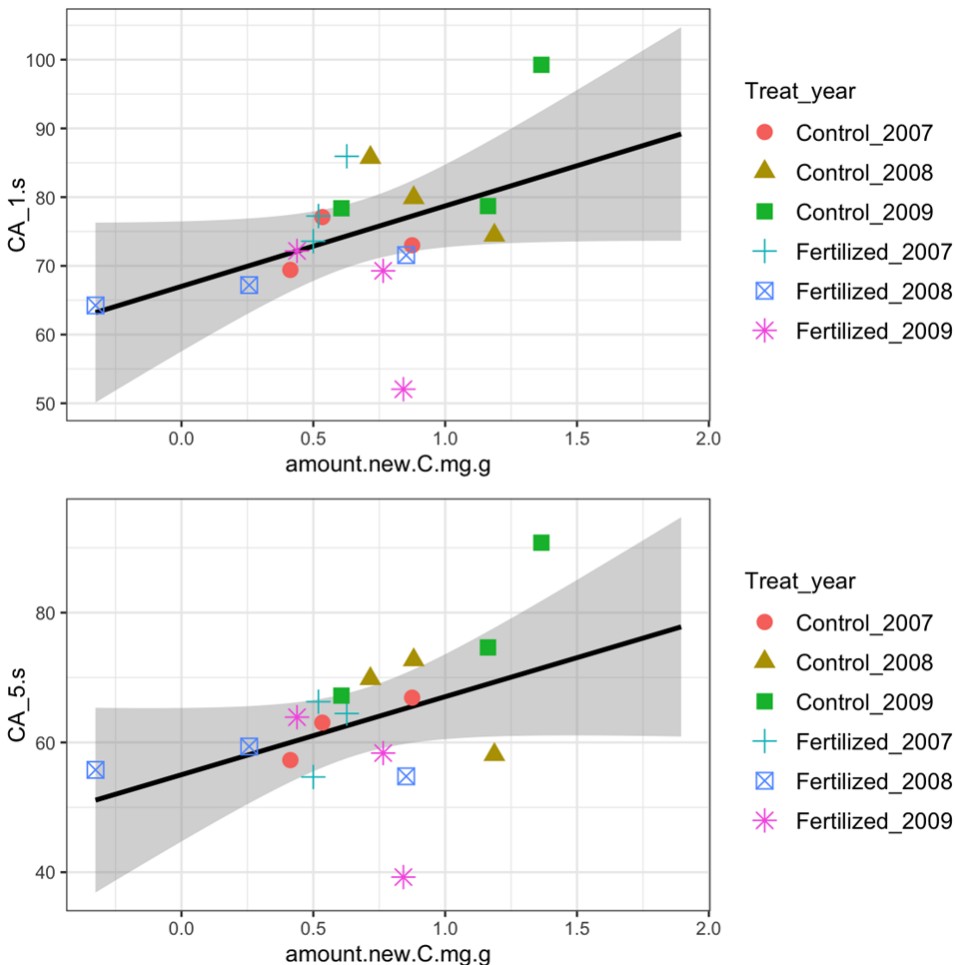


Figure 2: Correlation between the amount of new carbon in the meshbags and a) CA at 1s and b)
CA at 5 s.

The C/N ratio of the mesh bag content was 11.9 in the initial material, which increased to an
average of 14.6 and 12.2 after 3 years of incubation in the control and fertilized plots
respectively (Table 1). According to the two way ANOVA, fertilization had a significant



effect on the C/N ratios of the meshbags (ANOVA, F=6.1, p<0.05). The impact of incubation
time or the interaction between fertilization and incubation was not significant. During the
first two incubation years (2007, 2008) there were no differences between the C/N ratios in
the control and fertilized samples. During the third incubation year (2009) the C/N ratios in
the control samples were significantly higher than the C/N ratios in the fertilized samples.

**3.3 Overall effects of the treatments on total EMF and saprotophic fungi**
The total fungal EMF and saprotrophic fungal communities were significantly influenced by
incubation time and by fertilization according to the Permanova analysis (p<0.001; F=5.4 and
p<0.001 ; F = 8.4, respectively) (Fig 3a). Fertilization had no significant effect on the total
fungal community during the first year but during the second year and third year the
fertilization effect was found to be significant (pairwise Adonis, p = 0.06; F =2 and p = 0.02;
F = 5.3, respectively). The proportion of EMF sequences on total fungal sequences increased
significantly over time in the mesh bags from 11 % and 10% during the first growing period
in the control and fertilized plots, respectively, to 24% and 38% after two years of incubation,
and to 78% and 73% after three growing seasons (Fig 3b; Table 1).

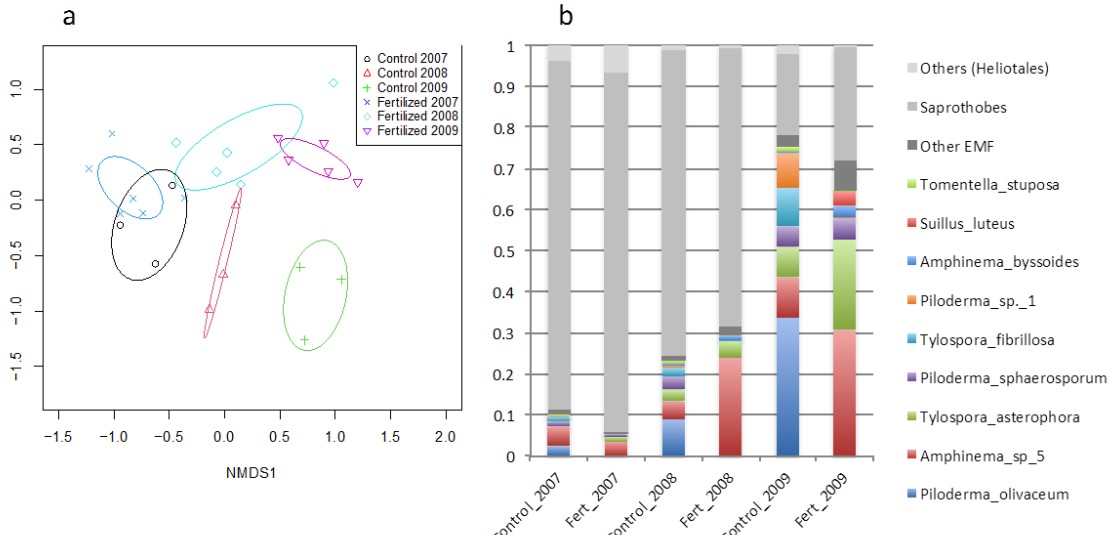


Figure 3:  Response of the fungal communities in the meshbags to the fertilization treatment and
incubation time. a) NMDS ordination analysis of the fungal communities b) Relative abundance of the
different fungal species.

## 3.4 Effects of fertilization on fungal community composition

After all bioinformatic processing and quality filtering, followed by rarefaction to a maximum
of 1200 sequence reads per sample (minimum 612), and elimination of all operation
taxonomic units (OTUs) that were only found in one sample, 26943 sequence reads were
recovered that were apportioned to 146 OTU's.
The total fungal communities (EMF and saprotrophic fungi) were significantly influenced by
incubation time and by fertilization according to the Permanova analysis ($p<0.001$; F=5.4 and
$p<0.001$ ; F = 8.4, respectively) (Fig 3a)
Fertilization had no significant effect on the total fungal community during the first year but
during the second year and third year the fertilization effect was found to be significant
(pairwise Adonis, $p = 0.06$; F =2 and $p = 0.02$; F = 5.3, respectively)



The proportion of EMF sequences increased significantly over time in the mesh bags (Dunn
test, $\chi^2$ =18, p<0.0001), (Fig 3b). 11 % and 7% of the sequences were EMF during the first
growing period in the control and fertilized plots respectively. These values increased to 24%
and 31% after two years of incubation in the control and fertilized plots respectively, and to
78% and 72%  after three growing seasons in the control and fertilized plots respectively
(Table 1).
Fertilization had a strong effect on all three dominant EMF genera (*Amphinema, Piloderma*
and *Tylospora*) (Fig 3b). *P. olivaceum* increased in abundance over time in the control plots to
become the dominating species (33% of the relative abundance) after three years of
incubation. This species was reduced to 0% in the fertilized plots independent of incubation
time (Fig 3b) *Tylospora fibrillosa* was also reduced in response to fertilization (Dunn test, $\chi^2$
=13.4, p<0.0001), (Fig 3b), while *T. asterophora* showed an opposite trend (Dunn test, $\chi^2$
=4.4, p<0.05).  Abundance of *Amphinema* was enhanced by fertilization and this species was
the most abundant one in the fertilized plots (Dunn test, $\chi^2$ =3.8, p<0.05) (Fig 3b)
**3. 5 Principal component analysis**
The principal component analysis separated the samples by incubation time along the
principal component 1. This component explained 34% of the variance.  Samples belonging
to the three-years incubation bags were ordinated to the right of the principal component 2
(Fig 4). Along the principal component 2 the samples were separated by the fertilization
treatment. This component explained 25.7 % of the variance. Samples belonging to the
unfertilized controls were ordinated above the principal component 1 (Fig 4). The linear
model showed that the fertilization/incubation treatments were significantly associated with
the PC1  (F=8.3, p < 0.01) and the PC2 (F=18 , p< 0.0001). The proportion of EMF to total
Basidiomycetes reads was strongly increased over time while the proportion of saprotrophic
fungi and Ascomycetes decreased with increasing incubation time. The EMF species
*Tylospora fibrillosa* and *Piloderma olivaceum* were positively related with the CA (initial
CA, $CA_{1s}$, $CA_{5s}$), with the C/N ratios and with the amount of new carbon inside the
meshbags; and their vectors were directed towards longer incubation time and opposite to the
fertilization treatments (Fig 4).

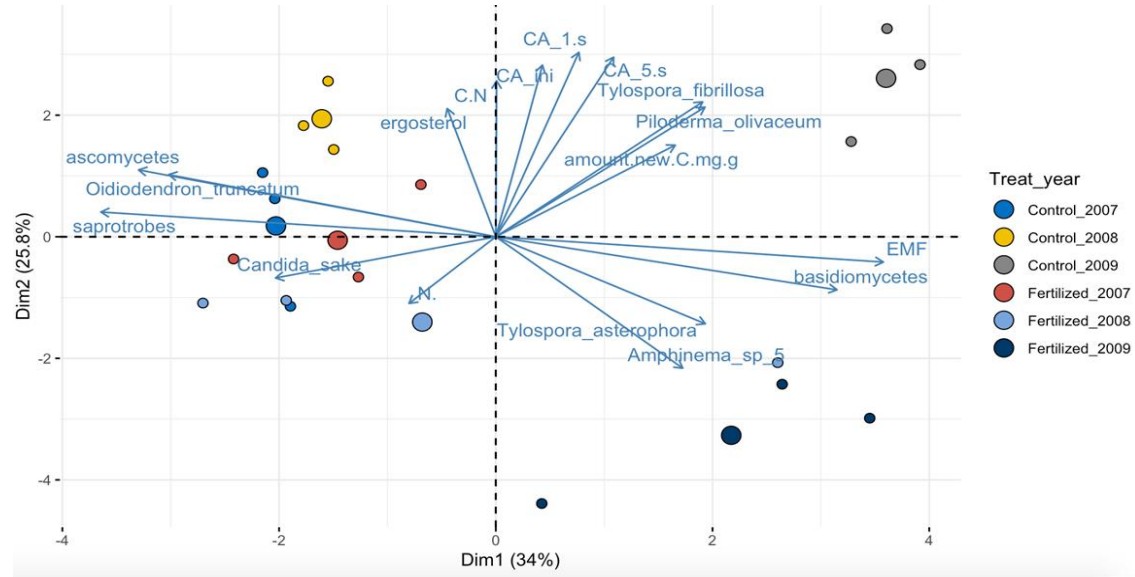


342       Figure 4: Principal component analysis of the most abundant fungal species and the properties of
the organic material inside the meshbags.


**4 Discussion**
**4.1 Effect of incubation and fertilization on the total fungal communities**
As expected, the fungal communities were influenced by the fertilization and by incubation
time and there was a significant increase in the percentage of EMF reads over time. It should,
however, be noted that the ingrowth of EMF in relation to other fungal groups was
surprisingly low during the first growing season (<12% of the fungal sequences), which is
much lower than what has been found in earlier studies (Parrent and Vilgalys, 2007;





Wallander et al., 2010). Some of this variation may be due to different weather conditions, the
first year was wetter than normal while the third was close to normal in precipitation
(Wallander et al., 2011), or due to larger belowground carbon allocation when the trees
approach canopy closure during the third year, as discussed in Wallander et al. (2010).
Therefore, we cannot ensure whether the difference of EMF reads between the incubation
periods responded to year-to-year fungal succession or if it was caused by the variation in
environmental conditions between individual years. Irrespective of the causes, the EMF
abundance was highest during the third year and this increase was associated with a higher
hydrophobicity, higher input of new C and higher C/N ratios suggesting a strong relation
between EMF and the changes in the surface properties of the organic material in the
meshbags.

364        The most dominant EMF genera in our study were *Amphinema, Piloderma* and

*Tylsopora* which also are common in other studies of EMF communities in coniferous forests
(Almeida et al., 2019; Walker et al., 2014; Tedersoo et al., 2008). In the control plots, the
most dominant species was *P. olivaceum* which did not colonize the meshbags collected from
fertilized plots. *Piloderma* is a common genus in boreal forests and is reported to be more
abundant in soils rich in organic N (Heinonsalo et al., 2015; Lilleskov et al., 2002), and to
decline in response to inorganic N fertilization (Teste et al., 2012), and elevated N deposition
(Kjöller et al., 2012; Lilleskov et al., 2011; Lilleskov et al., 2002a ; Taylor et al., 2000). The
decline of *Piloderma* in the fertilized plots in the present study is not surprising since this
genus produces abundant hydrophobic rhizomorphs that might constitute a large C cost for
the host (Defrenne et al., 2019), which is not economical for the symbiosis at high mineral N
concentrations. The increase in the C/N ratios of the meshbag substrates from the control
treatment might be thus an effect of biomass accumulation of *Piloderma* species, since EMF
fungi in general have a higher C/N ratio than maize compost (Wallander et al., 2003).



Additionally, it has been shown that *P. olivaceum* produces proteases that improve the ability
of the host trees to utilize N from organic compounds (Heinonsalo et al., 2015). Therefore, N
released from the maize compost by this fungus could have been transferred to the host plants,
which would contribute to the increase in C/N ratios in the control plots in comparison with
the fertilized plots. This explanation is consistent with results described by Nicolas et al.
(2017), who used FTIR and NEXAFS to analyze chemical changes of similar maize compost
incubated in mesh bags over one growing season in a Norway spruce forest in southwestern
Sweden. They found that heterocyclic-N compounds declined in mesh bags in comparison
with non-incubated reference material, which was interpreted as an effect of removal by EMF
and transfer to the host trees. This decline was higher in the unfertilized control plots
compared with fertilized plots. In the fertilized plots of the present study, the amount of new
C tended to increase in the three-year incubation bags where the C/N ratios reached the lowest
values, indicating limited N removal by the EMF colonizing these bags.
*Amphinema* sp. 5 responded positively to fertilization in our study which is supported by a
study by Kranabetter (2009) who found strong increase in the abundance of *Amphinema*
colonized root tips along productivity gradients in Canada. While a reduced abundance of *T.*
*fibrillosa* was observed in the fertilized plots, *T. asterophora* responded positively. Similarly
contrasting effects between this two species were found in other studies as well (Teste et al.,
2012 ; Kjöller et al., 2012; Toljander et al., 2006). In an N deposition gradient Kjøller et al.,
(2012) found increased abundance of *Tylospora asterophora* in areas with high N throughfall
while *T. fibrillosa* abundance decreased with higher N deposition.  Reduction of *T. fibrillosa*
in response to fertilization may be a result of C starvation since it has been shown that this
species is more dependent on C transferred from a living host in order to colonize new
seedlings on a clear cut compared to the more N tolerant *Amphinema* sp. which readily
colonized saplings on clear cuts (Walker and Jones, 2013).




**4.2 Effect of incubation and fertilization on hydrophobicity**

As expected, hydrophobicity increased over time in respect to the reference material
(non-incubated maize-sand mixture), and this increase occurred only in the unfertilized
controls. This increase in hydrophobicity was expected to be an effect of the accumulation of
fungal biomass and necromass over time as it has been shown that organic C (Woche et al.,
2017; Mataix-Solera & Doerr, 2004; Chenu et al., 2000) and microbial biomass and
necromass contribute to the hydrophobicity of soils (Schurig et al., 2013; Šimon *et al.,* 2009;
Capriel, 1997). However, the total amount of C was similar for all the incubation times and
was not affected by fertilization indicating that C content alone could not explain the
variations in hydrophobicity. Instead, the amount of new C entering the meshbags from
outside was found to be significantly correlated with hydrophobicity ($CA_{1s}$ and $CA_{5s}$). This
new C is expected to be of EMF origin as discussed by Wallander et al. (2011). Since
saprotrophic fungi utilize the maize compost material as their C source, it is expected that new
C inputs come from plant photoassimilates and are brought by EMF fungi (Wallander et al.,
2011). Therefore, these results suggest that the accumulation of biomass and necromass of
EMF origin over time might contribute to the buildup of hydrophobicity in SOM.

Our results show that fertilization reduced ergosterol concentration in the meshbags in
comparison with the control samples (Wallander et al., 2011) and this coincided with a
decrease in the hydrophobicity over time in comparison with the unfertilized controls and the
non-incubated reference material. It has been shown that fungi may enhance soil water
repellency of soil particles since some filamentous fungi produce insoluble substances like
ergosterol and hydrophobins (Mao et al., 2019; Rillig et al., 2010). For instance, Hallet et al.
(2001) found that soil hydrophobicity decreased when fungi were killed after fungicide



additions. Therefore, it is possible that the lower fungal biomass in the fertilized plots in our
study led to a decrease in hydrophobicity as incubation time in the soil increased. However
the concentration of ergosterol in the meshbags from the control plots did not increase with
incubation time and even tended to decline in the last incubation sampling when
hydrophobicity increased, indicating that ergosterol alone is not a good predictor of
hydrophobicity. It is possible that high ergosterol values after one growing season was an
effect of high abundance of yeast like *Guehomyces*, *Cryptococcus*, *Rhodotorula* and *Candida*,
which are unlikely to contribute to hydrophobicity but dominated the fungal communities of
the mesh bags during the first growing seasons. These fungi decreased drastically in
abundance in the three-years incubation bags. The ergosterol content per dry mass of yeasts
are much higher than in filamentous fungi (Pasanen et al., 1999), which might explain the
high ergosterol values in the first incubation periods. From these results we conclude that
hydrophobicity is more associated with EMF fungal colonization (measured as the amount of
new C) than with total fungal biomass (measured by ergosterol).
Given the apparent association of EMF colonization with higher hydrophobicity over time,
some EMF species may be expected to be more important than others for this process. We
expected higher hydrophobicity in the control plots in response to a higher proportion of
hydrophobic long distance exploration types species. The presence of *Piloderma* species like
*P.olivaceum,* known to form hydrophobic mycelia,  (Lilleskov et al., 2011, Agerer, 2001), and
that was totally absent in the fertilized plots is likely to contribute significantly to
hydrophobicity of SOM. On the other hand, the increase of *Amphinema* sp. 5, in the fertilized
plots which is also reported to form hydrophobic mycelia (Lílleskov et al., 2011), was not
accompanied by an enhanced amount of new carbon in comparison with the controls, which
may suggest that necromass from this fungus do not accumulate to the same extent as for
*Piloderma*, and is probably not associated with the hydrophobicity in the meshbags. These



findings suggest that hydrophobicity of living mycelium might not necessary influence the
water retention of the organic material to a large extent. This is consistent with the findings of
Zheng et al. (2014) who found that the hydrophobicity of EMF mycelium do not necessary
enhance soil water repellency. They tested how different EMF strains inoculated on *Pinus*
*sylvestris* affected water repellency of sandy loamy soil. The mycelium hydrophobicity of the
fungal strains used in their experiment was previously tested by drop immersion on fungal
mycelium growing on pure cultures. The authors found that the mycelium from hydrophobic
species generally enhanced water repellency but not all hydrophobic isolates had positive
effect on soil hydrophobicity.  It was suggested that beside mycelium hydrophobicity other
species-dependent factors like growth patterns, the degree of soil particles coverage or the
amount of hydrophobic substances produced by the fungus might influence soil water
repellency.  In the present study the difference in hydrophobicity between treatments might
not be related only to the exploration types of the abundant species but also by species-
dependent features. For example, the characteristic color yellow of *Piloderma* comes from an
insoluble pigment called corticrocin (Gray & Kernaghan 2020; Schreiner et al., 1997).
Moreover, the hyphae of *Piloderma* is reported to be coated with calcium oxalate crystals
(Arocena et al., 2001) probably as a strategy against grazers or repel water to avoid microbial
predation (Gray & Kernaghan 2020; Whitney  & Arnott 1987). Thus, these particular features
of *Piloderma* make it a good candidate to explain the enhanced the hydrophobicity of the
material in the control meshbags which is supported by the association between the
abundance of this fungus, the new C in the meshbags and the CA.

**4.3 Ecological significance**
The effect of fertilization on fungal communities and its significance for C sequestration has
been largely discussed (see Jörgenssen et al., 2021; Almeida et al., 2019; Högberg et al.,



2010; Janssens et al., 2010; Treseder, 2004). Additions of inorganic N may have a strong
positive effect on plant net primary production (Binkley & Högberg, 2016) but have also been
shown to decrease belowground C allocation (Högberg et al., 2010) and consequently
decrease EMF biomass (Almeida et al., 2019; Bahr et al., 2015; Högberg et al., 2007, 2010;
Nilsson & Wallander, 2003), which will reduce the input of C to the soils and may reduce C
sequestration. However, Bödeker et al. (2014) for example, showed that addition of inorganic
N significantly decreased the abundance of *Cortinarius acutus*, a species that can enhance
SOM decomposition in order to uptake N (Lindahl et al., 2021). The decrease of *Cortinarius*
sp was accompanied by a decrease in the enzymatic oxidation in the humus layer of the soil.
Therefore, it has been suggested that fertilization might improve C sequestration by
suppressing SOM decomposition by some key species EMF like *Cortinarius* (Lindahl &
Tunlid, 2015 ; Bödeker et al., 2014). In the current study we show that *Piloderma*, another
common species from northern-forested ecosystems, is negatively affected by fertilization and
that its decrease might be associated with a decrease in the organic material hydrophobicity.
These findings suggest that even if fertilization could reduce the abundance of EMF with
decomposer capabilities it may also reduce the accumulation of hydrophobic fungal mycelium
that could enhance SOM formation and C sequestration rates. Therefore, the role of different
abundant EMF genera like *Piloderma* and *Cortinarius* in boreal forests for establishment and
destruction of hydrophobicity and the effect of fertilization on them warrants further research.



**Author contributions:**

JPA: Conceptualization of the research goals and aims. Data curation and analysis.

Manuscript writing.

NR: Data acquisition, curation and analysis.

SW: Data acquisition, curation and analysis.

GG: Conceptualization and development of the methodology.

HW: Conceptualization and development of the methodology, research goals and aims.

Manuscript writing.

**Acknowledgments:**

This research was supported by a grant from FORMAS nr: 2018-00634 for H. Wallander.

**This manuscript is intended to the special issue: Biotic and abiotic feedback to climate change and implications for biogeochemical cycling in terrestrial ecosystems.**

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
