# Peer review of "Nitrophobic ectomycorrhizal fungi are associated with enhanced hydrophobicity of soil organic matter in a Norway spruce forest."

_Biogeosciences, 2022_

## Author Response (AR1)

**Mark Anthony´s comments (Reviewer 1)**

Thanks Mark for your positive comments! It is very nice to know that there are fellow young researchers interested in EMF. Organisms so interesting and so important for the biogeochemical cycles!

Even if minor, your comments improved the manuscript a lot.

**Comment:**

Line 56: change 'to' to 'for'

**Answer:**

Done

**Comment:**

Line 57: I have been curious of this framing of these results because ericoid mycorrhizal fungi include many very strong decomposers (e.g. Burke and Cairney 2002, Mycorrhiza; Kohler et al. 2015, Nature Genetics)

**Answer:**

That is true.

This statement is based on findings from Clemmensen et al. (2015) from boreal forests. As you mentioned Ericoid mycorrhiza could have decomposing abilities also and it would be interesting to mention more about this fungal guild in this introduction. However, the purpose of this sentence was to explain that fungal community composition can affect carbon stocks. Our paper explores EMF mostly so going more in detail into the ericoid mycorrhiza would feel a bit out of our scope.

**Comment:**

Line 61: Though a great study, I would not say that Lindahl et al. (2021) could conclude causality in their work, and thus I would not say that C. acutes 'resulted in...'. Rather, I would say 'was linked to'.

**Answer:**

Agree and done

**Comment:**

Line 65-68: These results by Lilleskov et al. (2011) are very important, but some of the summaries at the genus level need to be reconsidered. For example, increasing evidence suggests that members of the Russula and Lactarius genera include species that respond both negatively and positively to N additions (e.g. Morrison et al. 2016, Fungal Ecology; Van der Linde et al. 2018, Nature; Moore et al. 2021, GCB; Anthony et al. 2021, ELEMENTA).

**Answer:**

Russula and Lactarius were omitted

**Comment:**

Line 94-95: Each fertilisation treatments includes a 50 kg N ha-1 yr-1 range, why is this? Maybe adding one additional clause to clarify. Because you ultimately consider fertilisation a single treatment, it is easily defendable but it would be good to stand lone in the paper versus needing to read Bergh et al. (2008) first. Can you also describe how long the N addition treatment was fertilized prior to installing the mesh bags?

**Answer:**

The aim of the additions was to optimize growth without inducing N leaching, as explained in Bergh et al (2008). The following sentence was added to the text:

*In the fertilization treatments specific amounts of N (ammonium and nitrate) were applied to optimize plant growth without inducing leaching. The amount of N additions were based on needle N determinations and monitoring of N in soil water (Bergh et al 2008). Thus, the fertilization was applied by hand as 50-100 kg N ha$^{-1}$ every year for the first fertilization regime and as 100-150 kg N ha$^{-1}$ every second year in the second fertilization regime* **(fertilization begun in 2002)**.

**Comment:**

Line 95-96: Can you provide more information on the fertilisation of other macro- and micronutrients? Because micronutrient loss is also hypothesised to influence how fungi respond to N additions (e.g. Whalen et al. 2018, GCB).

**Answer:** The following text was added:

*To avoid nutrient imbalance caused by fertilization, the amount of micronutrients was adjusted to optimum nutrient proportions for Picea abies (as calculated by Ingestad 1978).*

**Comment:**

Line 116: It would be interesting to know why in November versus a time when tree growth and belowground C allocation is presumably higher.

**Answer:**

Different seasonal peaks for EMF biomass have been reported. Most commonly, EMF growth peaks in early Autumn probably due to more C allocated by the trees after the resource has been given to the leaves in Spring and Summer (This coincides with the production of fruiting bodies). However, it has also been reported that during warmer months EMF growth can peak. By collecting the samples in November, we can be sure that most of the productive season was captured. The one-year bags were incubated for 8 months so the bags were buried from Spring until late autumn. Therefore, independent of the seasonal peak the bags were belowground when more growth can be expected.

**Comment:**

*Line 154: Please add one sentence about how sequences were denoised, given this is 454 data it is especially important bioinformatic detail.*

**Answer:**

The following sentence was added:

*The trim seqs operation was run with the following exclusion parameters: all sequences that mismatched the sample ID barcode at more than one position, mismatched the primers at more than 2 positions, had homopolymers longer than 10 bp, were shorter than 150 bp, or had an average base call quality score below 20 over a moveable window of 40 bases.*

*Line 169-172: Was this part of the work done manually?*

**Answer:**

Yes

*Lien 174: I realise the bioinformatics was done many years ago, but because the submission is current, I encourage updating language around the Zygomycota to be consistent with current taxonomic consensus (see Spatafora et al. 2016, Mycologia).*

**Answer:**

Fixed

**Comment:**

*Line 175: Does 'unknown ectomycorrhizal status' also refer to non-ectomycorrhizal taxa or just taxa thought to be ecto but not confirmed?*

**Answer:**

It refers to taxa that are found in genera of ambiguous ectomycorrhizal status, either because their mycorrhizal status is unknown, or because a few members of the genus have been shown to be ectomycorrhizal under certain circumstances but the genus is generally not considered obligately ectomycorrhizal. This includes many ericoid and saprotrophic as well as ectomycorrhizal fungi. In our study this group was primarily populated by OTU's within the Heliotales.

**Comment:**

*Line 180-182: Maybe just say 'relative abundance'?*

**Answer:**

Done

**Comment:**

*Line 215: Can you also provide technical details on the C/N measurements and the ergosterol analysis?*

**Answer:**

The C/N ratios were calculated just by dividing total C by total N. Those were extracted using an elemental analyzer connected to an Isoprime isotope-ratio mass spectrometer (Isoprime, Manchester, UK).

Extra information of the basics of ergosterol extraction is added:

*To estimate ectomycorrhizal growth, the fungal cell membrane compound ergosterol was measured as a biomarker of fungal biomass. Ergosterol was extracted from 5 g of the pooled sand-maize mixture from the meshbag. Briefly the sample was subjected to saponification using a solution of 10 % KOH in methanol and the non-polar phase (where the ergosterol is present) was separated using cyclohexane. The ergosterol was quantified by high-performance liquid chromatograph (Hitachi model L2130), a UV detector (Hitachi model L2400). For more detailed regarding the protocol see Wallander et al. (2011)*

**Comment:**

*Line 221: Was this a PCA of all these variables together or was some type of vector fitting used to fit the non-fungal values (e.g. envfit function in vegan)? I am also a bit concerned of using PCA on relative abundance data given how many zeros are probably in the data and co-linearity among some of the taxa. Is there a reason why PCoA was not used with Bray-Curtis distance or a distance-based redundancy analysis also using Bray-Curtis dissimilarity? Both of these would be more suitable non-parametric alternatives. I am also guessing from Figure 4 that this is a PCA of both fungal relative abundances and organic matter properties together. Thus, was some type of transformation used to put everything on the same scale? Additionally, can you define what was the criteria for being the most abundant fungal OTU. Was there a cutoff based on sequence proportion and/or occurrence across the sampling units? I am also guessing this is at the OTU level?*

**Answer:**

No vector fitting was used in this PCA. This graph is not a plot from Vegan and it is not based on distance matrixes obtained from the relative abundance of the fungal community data. In this PCA I plotted the measurements of the organic matter properties of the meshbag contents and the Relative abundance of the most abundant EMF species. Therefore, the data was transformed to be in the same scale. The function PCA in R scales the vectors as a default function:

For this graph the criteria was the more abundant EMF species relative to the total fungal reads.  The most abundant genera in the control plots was *Piloderma* with up to 50% of the total fungal reads and *Piloderma olivaceum* alone contributed with about 30% of the reads by the third year. The same goes for the fertilized plots where the genus *Amphinema* contributed with up to 40% of the total fungal reads while *Amphinema* sp 5. contributed with up to 30% of the total fungal reads. The other species that do not belong to the genera *Piloderma, Amphinema* or *Tylopora* contributed with less than 5% of the total reads.

For the bar graph presented in figure 3 the species that contribute with at least 1% of the total fungal reds are presented individually. The others are grouped in the bar called "Other EMF".

**Comment:**

*Line 257: I think the alphabetic labels are missing on the figure (e.g., a, b, & c).*

**Answer:**
Fixed.

*Line 268: I would also be interesting to have the Pearson correlation coefficient here.*

**Answer:**

Done.

**Comment:**

*Line 271: I think Figure 2 could be cleaned up a bit so the legend and axis labels look nicer and do not contain underscores.*

**Answer:**

Fair enough! The figure has improved: Note that new panels have been added after a comment of another reviewer:
***When breaking down the data by fertilization regime, there was a positive correlation between amount of new C and the hydrophobicity for the CA5s in the control plots (Fig 2c) but the correlation tended to be negative in the fertilized plots (Fig 2d)***

[Figure]

**Comment:**

*Line 285: Why is it total fungal EMF and saprotrophic fungal communities? Is this because all other trophic groups and non-assigned to trophic group fungi were removed? Did you also look at EMF alone and saprotrophs alone?*

**Answer:**

The way it is written is indeed confusing. We did not exclude any data in this analysis. We used all the fungal guilds present in the communities. I change that part to:

*The total fungal communities were significantly influenced…*

**Comment:**

*Line 301-304: These details seem like they should be the first part of section 3.4, where the molecular results are first introduced.*
*Line 305-310: Already stated verbatim in section 3.3?*

**Answer:**

Changed! I remove one of the repeated sections. Thank you for noticing this repetition of the results. It´s peculiar… neither of the co-authors nor the other reviewers noticed it!!

**Comment:**

*Line 311-316: Already stated at lines 290-294.*

**Answer:**

Changed! Thank you for noticing this repetition of the results

**Comment:**

*Line 321. Missing period after '(Fig 3b)'*

**Answer:**

Fixed.

**Comment:**

*Line 331-333: What linear model was used? Please add these details into the methods section.*

**Answer:**

The method ***lm: Fitting Linear Models*** in the package stats in R uses the QR decomposition:

Sharma, A., Paliwal, K. K., Imoto, S., & Miyano, S. (2013). Principal component analysis using QR decomposition. International Journal of Machine Learning and Cybernetics, 4(6), 679-683.

**Comment:**

*Line 333-334: I do not believe you mean to say the 'proportion of EMF to total Basidiomycota', as it sounds like you calculated a ratio of the two, but rather, I think you mean to to say, 'the proportion of EMF and the proportion of EMF increased over-time…'. I would also write Basidiomycota and Ascomycota versus 'mycetes'*

**Answer:**

Fixed.

**Comment:**

*Section 3.5: You only need to site Fig. 4 one time.*

**Answer:**

Fixed.

**Comment:**

*Line 358: The idea of fungal succession is important, but it need not be independent from changes in environmental conditions across time. I personally do not feel this qualification is necessary here. It derails the momentum of your story so early on! If you feel it is essential to already provide a caveat in this first paragraph of the discussion, I encourage you make one that does not derail the traction of the main conclusion of this work. You could say something more powerful like: 'Whether shifts in EMF were due to selection of later succession fungal taxa as the forest aged versus variation in climatic conditions remains unclear, but is ultimately not particularly important in terms of understanding how shifts in EMF relate to soil organic matter cycling'.*

**Answer:**

I added the sentence suggested.

**Comment:**

*Line 394: I believe this result is robust around T. asterphora responding positively to N additions, but I am willing to hedge it is geographically unique. Van der Linde et al. (2018. Nature) suggest an N depo optimum around 9 kg N ha yr-1 for this taxon, which is quite low for many parts of Central Europe where the work was conducted. Since your work was more northern and in a boreal forest, I bet the results are quite different from more southern, temperate forests. This is just a comment for consideration. I do not think there is need to comment on this in the text.*

**Answer:**

Interesting observation. I agree that it might reflect a geographical pattern. This study was done in Sweden where N deposition levels are not as high as central and southern Europe. In a forest more south than the present study, Almeida et al. (2019) found that *T. asterophora* tended to respond to N fertilization but it was less marked than in the present study. Probably, this species reaches an optimum but it can only tolerate so much N so it decreases as N deposition keeps increasing further south.

**Comment:**

*Line 401: Could you explain why the N tolerance component of this sentence helps to explain this result in little more detail?*

**Answer:**

It doesn't help to explain.

Previously we presented some references saying that Amphinena is N tolerant so in this particular sentence where we compared T. fibrillosa with Amphinena probably we wanted to remark it that the later is N tolerant. I have remove that word.

**Comment:**

*Line 404: You found the hydrophobicity did not change until the third year in the control mesh bags, which is also when proportions of EMF dramatically increased to make this group dominant. This supports the idea that the increase in hydrophobicity was due to EMF accumulation. It could be worth stating this also in this paragraph.*

**Answer:**

Agreed! We added text:

*As expected, hydrophobicity increased over time in respect to the reference material (non-incubated maize-sand mixture), and this increase occurred only in the unfertilized controls **at the last sampling when the fungal communities in the mesh bags were dominated by EMF.***

**Comment:**

*Line 421-441: Such an interesting paragraph! The explanation around yeasts and their ergosterol content is particularly compelling.*

**Answer:**

Thanks!! It is tricky with this method to measure fungal biomass since it is hard to discriminate between yeast and filamentous fungi. Luckily we had sequencing data to help us discuss that.

*Line 442-473: Another very interesting paragraph! Good ideas to explain the role of Piloderma species in soil C cycling.*

**Answer:**

Thanks! Lately I think I have found *Piloderma* when opening some new meshbags (amended with Fe-oxide; they seem to love it for some reason). There were thick yellow rhizomorphs that clumped together and were very hard to break apart and homogenize for the

extraction. In one of the papers where they extract the yellow pigment they actually discuss that it was hard to dissolve even in organic solvents. Indeed, I noticed that those yellow little balls did not dissolve when extracting the ergosterol of this new samples I had. I think that that could be one of the reasons we do not see a change in ergosterol over time in the current paper. Probably *Piloderma* is hard to break apart to have a successful extraction of its ergosterol. Unfortunately, this in anecdotical evidence and I cannot discuss it in the manuscript ☹

**Comment:**

*Figure 1. Is this the average contact angle at 1 and 5 s combined?*

**Answer:**

No it is not, what it is meant by average is the number of samples used to measure the contact angle (n=3). The legend is very misleading I reckon now. I have changed it:

*Contact angle (CA) comparisons between a) control treatment and reference material b) fertilization treatment and reference material and c) control and fertilized treatments. Bars represent standard deviation (n=3).*

**Comment:**

*Figure 2. It could help to provide a sentence describing what CA at 1 s and 5 s means; otherwise, the figure is a little tricky to interpret as a stand-alone component.*

**Answer:**

Some clarification to figure legend was added:

*Figure 2: Correlation between the amount of new C and the hydrophobicity of the meshbag contents measured as the contact angle (CA). The C.A was determined directly after placement of the water drop at 1 second (1s) a) and 5 seconds (5s) b, c, d).*

Some clarification was also added in the figure legend 1:

*Figure 1: Contact angle (CA) comparisons between a) control treatment and reference material b) fertilization treatment and reference material and c) control and fertilized treatments. Shown is the intial CA (ini), determined directly after placement of the water drop and CA determined 1 second (1s) and 5 seconds (5s) after placement of the water drop. Bars represent standard deviation (n=3).*

**Comment:**

*Figure 3b: It is hard to differentiate between Tomentella stuposa and Tylospora asterophora and then between Piloderma olivaceum and Amphinema byssoides and then between Suillus lutes and Amphinema sp 5. Could you select colors with more contrast?*

**Answer:**

Done. The graph has changed. Now it has different color codes and also information about the exploration type and hydrophobicity as suggested by another reviewer.

**Comment:**

*Figure 4: Are the vectors the coefficients of the linear combos of the initial variables used to make the PCA? I would also write OTU instead of species*.

**Answer:**

They are the initial variables.

Christopher Fernandez´ comments (Reviewer 2):

Thank you Christopher for your comments. They definitely improved the manuscript! Especially the suggestion of coding the EMF species according to their exploration type and hydrophobicity.

**Comment:**

*How do you tease apart the effects of fungal in-growth from hydrophobic compounds entering the bags from the surrounding litter/SOM (e.g. ingress of hydrophobic compounds via transport in water)? While I think the evidence presented in the manuscript strongly suggests that the former is probably the major driver, I don't think one can completely rule out the potential effects that the surrounding litter and SOM (and associated changes with the N treatment) may have on the substrate properties in the in-growth bags. I would suggest adding a few sentences in the discussion about plausible alternative mechanisms.*

**Answer:**

This is a very good point. I agree that we cannot rule out the possibility that soil solution entered the meshbags during the incubation inside the soil.

However there is some indication that EMF might explain the new C inside the bags since the number of EMF reads was significantly correlated with the new carbon in the meshbags (Pearson, T = 2.4, p < 0.05). **I have added this result (not mentioned before) in the manuscript.**

I added a couple of sentences in the discussion stating that:

*It should be also noted that we cannot rule out the possibility that other compounds from the soil entered the meshbags during the underground incubation. In soils, polymeric substances coming from SOM, root or microbial exudates can have hydrophobic properties (Vogelmann et al., 2013; Mataix et al., 2007). Hence, the hydrophobic changes in the material could be partly explained by other sources than EMF mycelium. However, the significant correlation between the new carbon in the bags and the EMF reads and the negative effect of fertilization on the C.A might suggest that hydrophobicity changes in the meshbag content are caused mainly by EMF.*

**Comment:**

*L3 For correct grammar change "fungi" to "fungal" OR just omit "symbiosis"*

**Answer:**

Done.

**Comment:**

*L37 revise to say "...this mycelium turns into necromass…"*

**Answer:**

Done.

**Comment:**

*L44 the authors might want to add hypothesized mechanisms behind differences in decomposition rates among hi and ho SOM here*

**Answer:**

I added a short explanation about it:

*SOM can be protected from decomposition in aggregates where hydrophobic coatings of mineral particles change the physical properties of the particles, reduce water films around them and limit water penetration inside the aggregates. This affects the mobility of microbial decomposers and enzymes from the soil solution and reduces organic matter decomposition.*

**Comment:**

*L53 change "saprophyte" to "saprotroph" for consistency (and a more widely accepted term)*

**Answer:**

Done.

**Comment:**

*L60-63 I would suggest explicitly stating that this particular species of Cortinarius has retained the enzymatic capacity to breakdown complex SOM in order to access nutrients.*

**Answer:**

I added:

*Certain species of EMF may have exceptional importance for organic matter degradation as the presence of Cortinarius acutus (which has retained the enzymatic capability to breakdown SOM to access nutrients) was linked to 33% lower C storage in the organic top soils in 359 investigated stands in boreal forests in Sweden (Lindahl et al., 2021).*

**Comment:**

*L64-68 species are mentioned but genera are given as the examples*

**Answer:**

Replaced the word species by EMF.

**Comment:**

*L64-68 I would add a sentence stating that for Russula and Lactarius there is quite a bit of variability in response to N fertilization at the species level*

**Answer:**

We removed these genera

**Comment:**

*L66 missing a ",", in front of Suillus*

**Answer:**

Fixed.

**Comment:**

*L92-96 Please provide what form the N fertilizer was*

**Answer:**

A couple of sentences were added:

*In the fertilization treatments specific amounts of N (ammonium and nitrate) were applied to optimize plant growth without inducing leaching. The amount of N additions were based on needle N determinations and monitoring of N in soil water (Bergh et al 2008).*

**Comment:**

*L174 The Zygomycota is no longer a recognized Phylum (now split into the Mucoromycota & Zoopagomycota; Spatafora et al. 2016)*

**Answer:**

I changed it in the text.

**Comment:**

*Figure 1. This is a matter of style but I feel the data could be presented in a different way that is more intuitive and impactful (e.g. boxplots?). The information is there, for me it just was not conveyed immediately*

**Answer:**

I personally like this type of graph. I would like to keep it like that.

**Comment:**

*Figure 2 I would be curious to see the N treatment treated as a covariate in an ANCOVA. Is the control slope steeper compared to the fertilized? Just looking at the plots it would appear so and may bolster the support for arguments made in the discussion about Ho and not Hi biomass that is contributing to SOM hydrophobicity.*

**Answer:**

I am not sure what it means to use the N treatment as a covariate. A covariate is a continuous variable and the ANCOVA is used to test the effects of an independent categorical variable on a dependent continuous variable controlling for the effect of a second continues variable ( the covariate). I guess I can apply an ANCOVA for the data presented in Figure 2. Then I would be testing the effect of Fertilization on the Contact angle using the amount of new C as a covariate??

However, I am not sure that a covariate analysis will be the best for our data and hypothesis.

Generally, the covariate is not very relevant for the study question and the test is used to find if the differences between the treatments are due to initial selection differences (the covariate). So the test removes the variance between treatments caused by the covariate (Miller and Chapman, 2001 & Jamieson, 2004).

However, the amount of new C should not be used as a mere covariate since it is the main dependent variable that we try to use to explain the changes in hydrophobicity in time and fertilization regime. In this case the amount of new C might be intimately associated with the treatments (as we hypothesized) and removing its effect in and ANCOVA would remove most of the variance between the treatments (Miller and Chapman, 2001).

Jamieson, J. (2004). Analysis of covariance (ANCOVA) with difference scores. International Journal of Psychophysiology, 52(3), 277-283.

Miller, G. A., & Chapman, J. P. (2001). Misunderstanding analysis of covariance. Journal of abnormal psychology, 110(1), 40.

Talking about the slopes of the treatments, yes, the slope of the control is not only steeper but the correlation tends to be negative if the fertilized plots are analysed separately:

[Figure]

This make sense considering that the control and fertilized plots have opposite trends regarding hydrophobicity and as it is mentioned in the discussion; the changes in hydrophobicity cannot be explained only by the amount of hydrophobic EMF (being the new

C used as a proxy for EMF contribution). It depends on features specific of certain genera in this case *Piloderma* that was extremely scarce in the fertilized plots.

Additional information regarding this has been added in the results:

*There was a positive correlation between the amount of new C and the hydrophobicity for both CA1s and CA5s (Pearson, T = 2 , p=0.06; T = 1.9, p=0.07 ; respectively) (Fig 2a and b respectively).* **When breaking down the data by fertilization regime, there was a positive correlation between amount of new C and the hydrophobicity for the CA5s in the control plots (Fig 2c) but the correlation tended to be negative in the fertilized plots (Fig 2d)**

Also, those extra 2 figures were added as panel 2c and 2d in Figure 2.

**Comment:**

Figure 3b. Maybe in the key you could add the hydrophobicity of each of the taxa based on genus level classifications in Agerer 2001 and Lilleskov et al. 2011 for those that are not immediately familiar? Additionally, a third panel with the relative abundance of the two hydrophobicity groupings could be added.

**Answer:**

A new Figure 3 where the most abundant species (at least 1% of the total fungal reads) are grouped by genus and by exploration type and hydrophobic properties of their mycelium will be presented:

[Figure]

Also, in the same figure a boxplot of the most abundant genera will be shown:

Reviewer 3 comments:

Thank you for the comments. The manuscript has improved a lot and with the suggested changes our findings are supported.

**Comment:**

*For instance, Almeida et al. could code for mycelial hydrophobicity and compare the relative sequence abundance of hydrophobic EMF taxa over time/across treatments. The effect of fertilization is particularly important given that the design of the study hinges on the idea that EMF with hydrophobic mycelia decline with fertilization*

**Answer**:

A new Figure 3 where the most abundant species (at least 1% of the total fungal reads) are grouped by genus and by exploration type and hydrophobic properties of their mycelium will be presented:

[Figure]

Also, in the same figure a new panel with a boxplot of the most abundant genera will be shown:

[Figure]

**Comment:**

*I would also like to see more information about the variability within the EMF community. For instance, instead of a stacked bar chart, Figure 3b could be broken out by treatment and each taxon could have an error bar.*

**Answer:**

In the third panel in the new Figure 3 it can be seen how well represented the genera are across samples. Especially for the genus *Piloderma* there is low variability between replicates as it can be seen in the box plots especially in the third-year bags.

**Comment:**

*Further, much of the discussion centers on the physiology and ecology of the most abundant species of EMF in the control plots (Piloderma oliviceum),but little information is available regarding how consistently this taxon shows up in the meshbags. If P. oliviceum is indeed abundant across most samples, this would strengthen the discussion of its role in potentially enhancing SOM hydrophobicity.*

**Answer:**

*P. olivaceum* is quite well represented across the different replicates in the control plots as shown by the standard bar error:

[Figure]

However, now the results will center more around the **genus *Piloderma*** which is also well represented in the control plots as explained in my previous comment.

**Comment:**

*Also, if possible, regressing the relative sequence abundance of hydrophobic EMF against the averaged contact angle of the substrate in an ANCOVA would strengthen the claim that EcMF mycelial hydrophobicity is imparting increased hydrophobicity to meshbag contents. If this emerges across N fertilization treatments, this would be particularly impactful.*

**Answer:**

An ANCOVA was performed and the abundance of hydrophobic species were not significant while the effect of treatment was:

ANOVA Table (type II tests)

| Effect | DFn | DFd | F | p | p<.05 | ges |
|---|---|---|---|---|---|---|
| 1 Long.distance.exploration.type | 1 | 15 | 0.538 | 0.475 | | 0.035 |
| 2 Treatment.1 | 1 | 15 | 6.150 | **0.025** | * | 0.291 |

This means that fertilization does have an effect on the contact angle but this cannot be explained by the proportion of hydrophobic EMF species. This fits with the discussion where we argue that the hydrophobicity is conferred by species specific features of *Piloderma* which is very scares in the fertilized plot. I will explain more on the matter in my next

answer. However, I am not sure that a covariate analysis will be the best for our data and hypothesis.

The ANCOVA is used to test the effect of an independent categorical variable (in this case Fertilization) on a dependent continuous variable (Contact angle) controlling for the effect of a second continuous variable, the covariate (Hydrophobic EMF).

Generally, the covariate is not very relevant for the study question and the test is used to find if the differences between the treatments are due to initial selection differences (the covariate). So the test removes the variance between treatments caused by the covariate (Miller and Chapman, 2001 & Jamieson, 2004)

However, hydrophobic EMF should not be used as a mere covariate since it is the main dependent variable that we try to use to explain the changes in hydrophobicity in time and fertilization regime. In this case Hydrophobic EMF might be intimately associated with the treatments (as we hypothesized) and removing its effect in and ANCOVA would remove most of the variance between the treatments (Miller and Chapman, 2001).

Jamieson, J. (2004). Analysis of covariance (ANCOVA) with difference scores. International Journal of Psychophysiology, 52(3), 277-283.

Miller, G. A., & Chapman, J. P. (2001). Misunderstanding analysis of covariance. Journal of abnormal psychology, 110(1), 40.

As suggested, regressions between the relative abundance of the **sum of reads from the hydrophobic EMF species** (those contributing at least with 1% of the total fungal reads) and the **averaged contact angle** has been performed and are explained in the next answer.

**Comment:**

*45-47: The hydrophobicity of living EMF mycelia is framed as a possible driver of SOM hydrophobicity here, but the subsequent analyses do not address how hydrophobic vs. hydrophilic EMF differ in abundance over the fertilization treatments. Is there a way to either change this framing, or address it with further analyses?*

**Answer:**

Indeed, there is a trend for higher amount of hydrophobic types in the controls in comparison with the fertilized plots in the third year and a trend for lower amount of hydrophilic types in the fertilized plots in comparisons with the controls in the third year, but this trend was not significant:

[Figure]

Also, the proportion of hydrophobic species in relation with hydrophilic species tended to be higher in the controls but this was not significant:

[Figure]

Moreover, if we make a correlation between the averaged contact angle (initial, 1s, 5s) and the abundance of hydrophobic types, there is no trend nor significance. This means that the combination of hydrophobic species might not explain **completely** the hydrophobic differences between the two treatments.

However, if we make a regression between the averaged contact angle and the abundance hydrophobic types using only the **control plots** the correlation is significant (**this figure will be added as an extra panel in Figure 3):**

[Figure]

**This graph will be presented as another panel in figure 3 along with the other results from the community data.**

This makes sense considering that both treatments have opposite trends regarding hydrophobicity (in the **fertilized plots** there was a decrease in hydrophobicity over time) while the relative abundance of hydrophobic EMF increased and at by the third year there were more abundant hydrophobic species than the hydrophilic ones even in the **fertilized plots**.

[Figure]

If the changes in hydrophobicity are explained by the fungal communities it is likely that the effect of fungi is related more with specific and unique differences in the community composition between **control and fertilization**.

The most clear and unique difference between both treatments is the presence of the *Piloderma* species which almost disappeared in the fertilized plots. It should be noted that by the third year the proportion of hydrophobic species in the control plots was up to 60% of the total fungal reads but the genera *Piloderma* alone contributed to up 50% of the total fungal reads in the three-years bags.

Therefore, this genus is a good candidate to explain the increase in hydrophobicity in the control plots.

This is already mentioned in the discussion; the changes in hydrophobicity cannot be explained only by the exploration type and the hydrophobic properties of the mycelium of the species. It depends on features specific of certain genera in this case *Piloderma*. This was tested before by Zheng et al. (2014) in an elegant experiment where they measure hydrophobicity of sandy soil. The fact that the long exploration types correlates with hydrophobicity only in the controls (where *Piloderma* makes up the majority of the reads) but not in the fertilized plots where *Piloderma* spp. is almost absent further supports the discussion of this paper.

I have added extra information concerning this in the results:

*The more abundant hydrophobic EMF genera were Amphinema and Piloderma while the more abundant hydrophilic genera were composed of Tylospora and Tomentella (Fig 3b). The amount of combined hydrophobic species tended to be higher in the control plots (up to 60% of the total fungal reads) in comparison with the fertilized plots (up to 50% of the total fungal reads) in the three-years-incubation bags, but this increase was not significant. Additionally, the proportion of hydrophobic EMF species in relation to hydrophilic EMF species in the control plots tended to be higher than in the fertilized plots in the three-years bags but this was not significant. When both treatments (control and fertilization) where analyzed together, there was no correlation between the proportion of hydrophobic species and the contact angle. The proportion of hydrophobic species was positively correlated with the averaged contact angle (initial C.A, C.A at 1s and C.A at 5s) in the control plots (Pearson, T = 2.9, p<0.04) but not in the fertilized plots.*

*Piloderma increased in abundance over time in the control plots to become the dominating genus (up to 50 % of the relative abundance) after three years of incubation. The most dominant species in the control plots was Piloderma olivaceum which was reduced to 0% in the fertilized plots independent of incubation time (Fig 3b). Tylospora fibrillosa was also reduced in response to fertilization (Dunn test, $\chi^2$ =13.4, p<0.0001), (Fig 3b), while T. asterophora showed an opposite trend (Dunn test, $\chi^2$ =4.4, p<0.05). Amphinema sp 5. was the most abundant species in the fertilized plots and was enhanced by fertilization (Dunn test, $\chi^2$ =3.8, p<0.05) (Fig 3b)*

I have added extra information concerning this in the discussion:

*Given the apparent association of EMF colonization with higher hydrophobicity over time, some EMF species may be expected to be more important than others for this process. We expected higher hydrophobicity in the control plots in response to a higher proportion of hydrophobic long distance exploration types species.* **Indeed, the proportion of hydrophobic EMF species in the control plots tended to be higher in comparison with the fertilized plots in the meshbags incubated for three years. From the hydrophobic species in the control plots, Piloderma spp. constituted the majority of fungal species with up to 50% of the total fungal reads.** *The presence of Piloderma species like P.olivaceum, known to form hydrophobic mycelia, (Lilleskov et al., 2011, Agerer, 2001), and that was totally absent in the fertilized plots is likely to contribute significantly to hydrophobicity of SOM.* **In the**

*fertilized plots there was also an increase over time in the amount of hydrophobic EMF species (Amphinema being the most abundant hydrophobic genus) which was not accompanied by an increase in hydrophobicity. This may suggest that necromass from Amphinema do not accumulate to the same extent as for Piloderma and is probably not associated with the hydrophobicity in the meshbags. These findings suggest that hydrophobicity of living mycelium might not necessary influence the water retention of the organic material to a large extent. This is consistent with the findings of Zheng et al. (2014) who found that the hydrophobicity of EMF mycelium does not necessary enhance soil water repellency.*

**Comment:**

*48-50: I'm unclear about how by removing N and P from SOM, EMF activity may reduce soil C stocks. By inhibiting saprotrophic activity by outcompeting them for nutrients, wouldn't this enhance soil C stocks by reducing respiration by saprotrophic fungi (Gadgil effect)? Do you mean that EMF themselves are mineralizing C from SOM, decreasing soil C stocks?*

**Answer:**

We agree that our wording here was not optimal. We have now reformulated the sentence to:"

*In contrast to carbon accumulating activities by EMF, certain species may also reduce soil C stocks by oxidizing organic matter to release nitrogen and phosphorus. Some EMF species use 'brown-rot' Fenton chemistry and some use 'white-rot' peroxidases to do decompose SOM (Shah et al., 2016; Lindahl and Tunlid, 2015; Bödecker et al., 2014). This can result in 30% decrease in SOM according to Lindahl et al (2021).*

**Comment:**

*81-83: Here, you write that you would expect higher hydrophobicity in the control vs. fertilized plots due to the higher proportion of hydrophobic species – where is this tested? Genus-level assignments on mycelial hydrophobicity are available in the literature, as well as exploration type assignments that could offer further resolution on the effect of EMF mycelial traits on substrate hydrophobicity.*

**Answer:**

See my previous answer from the comment for lines *45-47.*

**Comment:**

235: Table 1 would benefit from indicators of significance, either between treatments or over time. I found myself asking questions like "is the decline in the contact angle in the fertilized plots over time significant?" and struggling to locate the relevant information in the text. Alternatively, it could be visualized, which would also make it easier to interpret.

**Answer:**

Letters of significance were be added to the table:

Table 1:
Average and standard error of the ergosterol concentrations, total C%, C/N ratio, amount of new carbon (C3 mainly from EMF), % of EMF DNA reads, and contact angle determined 5 seconds after placement of water droplets placed on mesh bags material amended with maize compost ($CA_{5s}$; estimation of contact angle stability). **Low scores letters refer to statistical differences according to posthoc Tukey test and pairwise Dunn test. Asterisks correspond to statistic differences for the C.A after 5 (s) between the meshbag contents and the non-incubated reference material.**

258-260: Looks like you're missing figure labels (a, b, c).

**Answer:**

Fixed.

**Comment:**

*272-273: Figure 2, would it be possible to make this a little cleaner (axis titles, the legend title)?*

**Answer:**

Fixed. A better figure will be added. Note that new panels have been added after a comment of another reviewer:
***When breaking down the data by fertilization regime, there was a positive correlation between amount of new C and the hydrophobicity for the CA5s in the control plots (Fig 2c) but the correlation tended to be negative in the fertilized plots (Fig 2d)***

[Figure]

**Comment:**

*Also, when the proportion of EMF reads is so low during the first two harvests, how can you attribute new C (C3 ingrowth into C4 substrate) to EMF alone? Table 1 indicates that roughly half of the "new" carbon enters the substrate by the end of the first incubation, although the proportion of EMF reads is only ~ 10%.*

**Answer:**

That is a very good observation and indeed it looks peculiar that the amount of new C whose increase does not seem to match the increase in EMF sequences over time. The method used to measure the new carbon in the meshbag is not perfect it seems, and it comes with it´s flaws.

It seems to me that there is some extra carbon coming inside the meshbags that is not EMF. For example some soil solution might have come inside the bag during the first two years of incubation and contributed with some new C. Then the non-EMF new C might have built up so by the third year we do not see a clear difference between the 3 incubation periods.
So the method seems not to be perfect.

In that case it is hard to conclude that **all the new C** in the bags comes exclusively from EMF. However there is some indication that EMF might explain part of this new C since the number of EMF reads was significantly correlated with the new carbon in the meshbags (Pearson, T = 2.4, p < 0.05).

So, there is a trend for samples with high EMF sequences to have high new C. **I have added this result (not mentioned before) in the manuscript.**

An extra input of soil solution could be expected to affect both treatments, Fertilized and Control, and that could also explain why we do not see much difference between both treatments regarding new C.

Also, in Wallander 2011 (where part this data is published) a correlation between the new carbon and the ergosterol in the bags was found.

The positive correlation of EMF and the amount of new C have been added to the results and some extra information about other C than EMF coming inside the bags has been added to the discussion (as requested by referee number 2):

*It should be also noted that we cannot rule out the possibility that soil solution entered the meshbags during the underground incubation. In soils, polymeric substances coming from SOM, root or microbial exudates can have hydrophobic properties (Vogelmann et al., 2013; Mataix et al., 2007). Hence, the hydrophobic changes in the material could be partly explained by other sources than EMF mycelium. However, the significant correlation between the new carbon in the bags and the EMF reads and the negative effect of fertilization on the C.A might suggest that hydrophobicity changes in the meshbag content are caused mainly by EMF.*

**Comment:**

*296: Figure 3, could you break panel b out to provide more information about how variable the EMF community is? Also, could you provide visual information about how the traits of the EMF community (hydrophobicity and/or exploration type) may be shifting according to sequencing results?*

**Answer:**

 Done.

**Comment:**

*300: How do different kinds of EMF respond to the fertilization treatment over time? Hydrophobic vs. hydrophilic genera?*

**Answer:**

See my previous answer from the comment for lines *45-47.*

**Comment:**

*359-363: Here you argue that overall EMF abundance is linked with higher hydrophobicity. This is different from your original framing, where you implicate EMF producing hydrophobic mycelia.*

**Answer:**

The main point of this sentence was to stress that an increase of **EMF over time** (while **non-EMF decreased**) was associated with the changes in the properties of the meshbag contents like C/N, new C and hydrophobicity.

However, since the increase in hydrophobicity and C/N ratios occurred in the control plots I have changed this sentence to be more precise:

*Thus, the EMF abundance was highest during the third year and this increase was associated* **with higher C/N ratios and hydrophobicity in the control plots** *and higher input of new C in the control and fertilized plots. This suggests a strong relation between EMF and the changes in the properties of the organic material in the meshbags.*

Later in the discussion (in the section: 4.2 Effect of incubation and fertilization on hydrophobicity) I go more in detail about why and how EMF colonization could have helped building up hydrophobicity).

**Comment:**

*372-374: What is the mechanism of partner selection implied here? Reduced total C allocation to EMF fungi? How do you rule out environmental filtering and/or changes in EMF C sink strength? The Defrenne study cited here is correlative – how does it support your causal claim?*

**Answer:**

We agree that our data does not support causal explanations here. We modified the text to:

*The decline of Piloderma in the fertilized plots may be a result abundant hydrophobic rhizomorphs that constitute a large C cost for the host (Defrenne et al., 2019), which is not economical for the symbiosis at high mineral N concentrations. Other more direct effects of the N fertilizer on the growth of Piloderma mycelium is however also possible.*

**Comment:**

*375-376: This is where I think more information about the homogeneity vs. patchiness of the EMF community would be helpful – are Piloderma spp. abundant across all samples?*

**Answer:**

Yes. The genus *Piloderma* seems homogenously distributed in the Control plots replicates (in the fertilized plots this genus is extremely scarce) and the variability between samples in the control plots is not big (especially in the third year incubation) as you can see in this boxplot:

[Figure]

**Comment:**

*415-419: Refer to comment from line 235. How can you attribute new C to EMF when their relative abundance after the first incubation was so low?*

**Answer:**

See my previous comment on the matter.

**Comment:**

*Also, the synthesis offered here (EMF necromass and biomass may contribute to SOM hydrophobicity) deviates from your original hypothesis, that EMF with hydrophobic mycelia in particular are contributing to SOM hydrophobicity.*

**Answer:**

In this context I am discussing the correlation between the new carbon and the contact angle as a possible explanation for the building up of hydrophobicity in the meshbags.

However, since Hydrophobicity is enhanced over time only in the control plots, I added that specification in the discussion:

*Therefore, these results suggest that the accumulation of biomass and necromass of EMF origin over time might contribute to the buildup of hydrophobicity in SOM **in the control plots.***

Moreover, I added another specification in the results:

*There was a positive correlation between the amount of new C and the hydrophobicity for both CA1s and CA5s (Pearson, T = 2 , p=0.06; T = 1.9,  p=0.07 ; respectively). **When breaking down the data by fertilization regime, there was a positive correlation between amount of new C and the hydrophobicity for the CA5s in the control plots (Fig 2c) but the correlation tended to be negative in the fertilized plots (Fig 2d)***

It is not surprising that the new C is correlated with the contact angle in the control plots. It is true that the new C might come from EMF in general but as it is mentioned later in the discussion (and earlier in my answers to the comments), the hydrophobic species in the controls contribute with the majority of fungal reads so they must be the ones contributing mostly to the new C and the changes in the water repellency of the material.

Later in this section I discuss the role of the individual species with hydrophobic mycelium in the control plots.

This is how the narrative and the flow of the discussion has been built.  First, we see some indications of the effect of the new C on hydrophobicity and discus the role of EMF but at the end we argue for *Piloderma*. Is like a puzzle where the first pieces point out to an overall effect of EMF but ultimately is *Piloderma* that might explain the hydrophobicity changes in the material.

**Comment:**

*437-439: This argument suggests that filamentous fungi contribute more than yeasts to SOM hydrophobicity – that is a much larger group than EMF fungi. How can you parse between the effects of filamentous fungi at large and EMF?*

**Answer:**

Yes, filamentous fungi are not restricted to EMF only. However (as discussed in the first paragraph of the Discussion section 4.1) EMF became the most abundant fungal guild in the three-years-incubation bags with about 80% of the DNA reads in the Control plots. In the first years where non-EMF fungi were the most abundant fungal guild the new carbon, contact angles, and C/N ratios were the lowest which might suggest that the changes of the meshbag contents were more associated with EMF than with other filamentous fungi. That is why in this paragraph I argue that overall EMF abundance might be linked with higher hydrophobicity.

The reason we mentioned yeasts in this context was to attempt to explain the lack of an increase in **ergosterol** values over time while there **was** an increase in EMF reads and new carbon.

**Comment:**

*445-449: This discussion would be strengthened by a more robust quantitative analysis of the relative abundance of hydrophobic/hydrophilic EMF across treatments. Where is your final hypothesis addressed?*

**Answer**:

As mentioned in the comments above the regression between the averaged contact angle and the amount of combined hydrophobic species were performed and supported the hypothesis that hydrophobic EMF increased hydrophobicity in the control plots but not in the fertilized plots probably by the presence of the *Piloderma* genus which is absent from the fertilized plots.

Therefore, testing the correlation between CA and hydrophobic EMF with both treatments combined (control and fertilized) and then for each of them separately gives good support for the different hydrophobicity patterns between fertilized and control plots.

In addition, the multivariate analysis done as the PCA tested for all the different continuous variables measured in the organic material inside the meshbags and gave a good indication of how the **most abundant species** are associated with the different properties of the organic material.